



**A Multiscale Spatial Dataset for Policy-Driven Land Developability across the**
**United States, 2001—2011**
**Authors**
Hung Chak Ho[1], Guangqing Chi[2*]
**Affiliations**
[1] Department of Urban Planning and Design, The University of Hong Kong, Hong Kong
[2] Department of Agricultural Economics, Sociology, and Education, Population Research
Institute, and Social Science Research Institute, The Pennsylvania State University, USA
**Corresponding author:** Guangqing Chi, Department of Agricultural Economics, Sociology, and
Education, The Pennsylvania State University, University Park, PA, 16802, USA, gchi@psu.edu



**Abstract**
Land vulnerability and development can be restricted by both land policy and geophysical
limits. Land vulnerability and development cannot be simply quantified by land cover/use
change, because growth related to population dynamics is not horizontal. Particularly, time-
series data with a higher flexibility considering the ability of land to be developed should be
used to identify areas of spatiotemporal change. By considering the policy aspects of land
development, this approach will allow one to further identify the lands facing population
stress, socioeconomic burdens, and health risks. Here the concept of "land developability" is
expanded to include policy-driven factors and land vulnerability to better reconcile
developability with socio-environmental justice. The first phrase of policy-driven land
developability mapping is implemented in estimating land information across the contiguous
United States in 2001, 2006, and 2011. Multiscale data products for state-, county- and
census-tract-levels are provided from this estimation. The extension of this approach can be
applied to other countries with modifications for their specific scenarios. The data generated
from this work are available at https://doi.org/10.7910/DVN/AMZMWH (Chi and Ho, 2020).



## 1 Introduction

Land cover and land use data have been commonly used for urban development and

regional health planning (Abrantes et al., 2016; Gounaridis et al., 2018; Hedblom et al., 2017;

Sharaf et al., 2018). These datasets allow identifying the locations more suitable for land

development and can also be applied to analyze the influence of land use and development

on socioeconomic burdens and community health risks. However, these data are missing

legal and land policy information. Some land development is restricted by policy; for

example, to prevent the loss of ecological systems and/or cultural heritage (Chi, 2010).

Regional development-restricted land can influence the forecasting and estimation of

changing health risks as well as socioeconomic vulnerability over several years. Therefore, a

comprehensive land use dataset should include land policy in mapping to take both social

and environmental justice into account when estimating "land developability."

This approach is important for application in current and future decades. Facing

exponential population growth, global land resources cannot support and sustain local

communities (Giampietro, 2018). Therefore, there is always a debate as to whether a specific

land area is developable or vulnerable (Oberlack et al., 2016), including a social concern in

that population stress from land development has been a key challenge threatening local

populations (Chi and Ho, 2018). As such, incorporating land policies with regional planning

has become an alternative control on land development (Lyles et al., 2014; Trop, 2017), as

the effects of land polices on planning can ultimately change urban forms and choices of

locations for development. From an environmental perspective, land policies in sustainable

planning are to, at minimum, reserve a specific area for resource management and

conservation. This can minimize potential disasters predicted by the Malthusian theory of

population (Petersen, 1999). From a health perspective, policy-restricted lands have lower

eco-environmental vulnerability, and these regions provide lower adverse health effects to

surrounding areas.



It should therefore be concluded that better estimating land developability with an eye
toward both social and environmental justice is an alternative pathway that considers both
land developability and land vulnerability through land policy and legal matters. This is
particularly critical because all growth related to population dynamics is not horizontal.
There can be a large spatiotemporal variability of population across regions, while some
areas may have very low population growth due to land policies. As a result, change in health
burdens as well as socioeconomic problems through space and time can be vastly different
across regions. It is therefore necessary to consider the ability of land development with
greater flexibility. Particularly, multiple years of data can be used to identify areas of change
from prior decades to evaluate how the land development has been changed
spatiotemporally. This can be further used to identify where the population-stressed lands
are. In addition, the index can identify how areas and municipalities can adapt to stress by
combining with other datasets (e.g., socioeconomic data). Based on further analysis,
implications for the environment can be provided to expand the concept of developable
lands in a context of unintended consequences.
The first phase for estimation of land developability is conducted based on the land
information across the contiguous United States. Multiscale data products for state, county
and census-tract levels are provided from the estimation. The contiguous United States is
selected as our first study site because it represents a typical developed country; the results
be used to create similar datasets for other developed countries. The extension of such an
approach can be modified based on specific scenarios in both developed and developing
countries, with the goal of implementing the concept of land developability that can
ultimately achieve greater success for global sustainability and development.

**2 Methods**
2.1 Data parameters



The land developability of the United States each year is estimated from the results of spatial
multicriteria analysis (SMCA) and zonal statistics, with five data parameters: 1) surface
water, 2) steep slope, 3) built-up land, 4) wetland and protected wildlife area, and 5) tax-
exempt land.

Surface water—rivers, lakes, and oceans—is extremely unsuitable for land

development. Doing so can involve legal and practical hurdles (Albert et al., 2013), the need
for ecosystem protection and restoration (Harrison et al., 2016; Martinuzzi et al., 2014), and
the possibility of natural disasters (Imaizumi et al., 2015).

Steep slopes can be unpractical for development because of loose soils and a high

probability of natural hazards such as landslides (Imaizumi et al., 2015; Liu et al., 1994; Zhou
et al., 2015). Development on steep slopes may therefore result in property damage and loss
of human life (He and Beighley, 2008). Legal requirements, such as Wisconsin's Erosion
Control and Stormwater Management Ordinance of 2002, also restrict development on
these landforms (Chi, 2010).

Built-up land, especially when pervasive, produces a densely built environment that may

have high environmental risks caused by poor ventilation and lower air quality (Ng, 2009).
These areas may also include large percentages of socioeconomically disadvantaged
populations, resulting in higher community risks when the neighborhoods lack sustainable
policies for urban transformation (Ho et al., 2017).

Wetland is a major natural resource that can serve as a diverse ecosystem (de Groot et

al., 2012), carbon sink (Mitsch et al., 2013), and natural purifier of water and air pollution
(Zhang et al., 2012). The loss of wetland brings risks such as higher levels of soil erosion and
vulnerability to drought (Ockenden et al., 2014; Wright and Wimberly, 2013). Similar to
wetlands are regions that protect habitats for endangered or threatened species, and
provide for other activities  (Watson et al., 2014). Federal and state regulations and land
policies constrain land development in these areas (Chi, 2010).



Finally, tax-exempt land in the United States includes federal- and state-owned regions

that are legally protected and publicly owned, and are restricted from residential,
commercial, or other types of land development.

2.2 Spatial data processing
Surface water coverage in this study was based on information from the National Land Cover
Database (NLCD) for 2001, 2006, and 2011 (Homer et al., 2004, 2007, 2015). NLCD is a
satellite-based product of the Multi-Resolution Land Characteristics Consortium and the U.S.
Geological Survey (USGS) and has adopted a land use classification scheme of eight major
categories.

Surface water in our study is the "open water" subcategory under the "water" class in

NLCD, consisting of areas with less than 25% vegetation and soil coverage within a radius of
approximately 30 meters.

Steep slope is defined as all with a slope ≥20%, based on data retrieved from the Digital

Elevation Model (DEM) under the Shuttle Radar Topography Mission (SRTM). SRTM is an
international research program of the Consultative Group on International Agricultural
Research—Consortium for Spatial Information (CGIAR-CSI), which records global elevations
at a resolution of 3 arcseconds (Jarvis et al., 2008). The original data in this dataset were
collected in February 2000 from a specially modified radar system during an 11-day satellite
mission, and SRTM Version 4 is a hole-filled DEM that was modified from the original data
using  a method of void-filling interpolation (Reuter et al., 2007). Reclassification was applied
to the slope to spatially delineate the areas with gentle slopes (<20%) and steep slopes
(≥20%).

Built-up lands are areas (approximately 30 m radius) with 20% or more impervious

surfaces. They are identified based on NLCD. Built-up lands commonly contain single/multi-
family houses, apartments, townhouses, and other commercial/industrial land.



Wetland and protected wildlife areas were retrieved from the datasets mentioned
above, as well as from NLCD, the USGS Federal and Indian Lands map, and University of
California-Santa Barbara's Managed Areas Database (MAD). The Federal and Indian Lands
map contains information on tax-exempt federal and state lands and national and state
protection areas. MAD includes spatial information on federally and state-managed areas, as
well as Indian and military reservations (McGhie et al., 1996). The lands classified as wetland
in NLCD were "woody wetlands" and "emergent herbaceous wetlands." The USGS Federal
and Indian Lands map listed protected wildlife areas as "wilderness," "wilderness study
area," and "wildlife management area"; and wildlife areas in MAD were "wilderness,"
"wilderness study area," and "wild and scenic area."
Tax-exempt land was identified from the USGS Federal and Indian Lands map and MAD.
It included all federally or state owned areas (forests, parks, trails, wildlife refuges, fisheries)
that were retrieved from these datasets.

2.3 Geovisualization of land developability in multiple scales
SMCA is a statistical method that can combine spatial data layers. During analysis, each data
layer is assigned a specific weight that considers its importance in terms of risk or
vulnerability. To avoid subjectivity, as documented in the 2002 guidelines of the United
Nations Environment Programme (Ho et al., 2018), we used an additive approach, giving
equal weight to all spatial layers.
We applied SMCA to map land developability using the following procedure:
1)   Spatial data layers that represent the undevelopable lands defined previously were

resampled into binary layers in raster format. The resultant layers were at a 90 m

resolution, with 1 indicating an undevelopable area and 0 indicating a location that

is theoretically developable.

2)   All binary layers were overlaid, and the sum of all values from pixels at the same



location were calculated.

3)   The layers of sums of all values were reclassified by the following criteria: if a

location has a value ≥1, it was changed to 0 to indicate undeveloped land. If it was 0,

it was re-designated 100 to signify 100% land developability within a 90 m pixel.

We applied the zonal statistics  to the subsequent map in binary format to estimate the

percentage of land developability based on the boundary of each state, county, and census
tract. We repeated this estimation to calculate land developability at the state, county, and
census-tract level across the United States separately for 2001 and 2011.
All land developability maps were then launched to a web-based GIS platform through an
application programming interface (API) powered by the Environmental Systems Research
Institute (ESRI), with base maps provided by the ESRI.

**3 Results and Discussion**
3.1 Web GIS platform for geovisualization of land developability
The first phrase of this study is a launch of county-level land developability data across the
United States in 2001, 2006, and 2011 through a web GIS platform for geovisualization
(www.landdevelopability.org). Figures 1 through 3 show the spatial distribution of county-
level land developability. In general, metropoles along the East and West Coasts and the
urbanized areas near the Great Lakes have lower land developability. There is also a lot of
land with low developability in the Western part of the United States, possibly because of
restrictions on land development on Native American or federal lands. In comparison, rural
counties in the Midwest show the highest potential for land development, followed by the
rural counties in the Northeast and South. Visually comparing the maps of 2001, 2006, and
2011, the land developability in the rural counties in the Northeast and the South has



significantly dropped over the years, while the potential for land development in the
Midwest counties has decreased, but generally not as fast.

3.2 Technical validation
Because this index is developed in a qualitative-based context, we first apply a detailed
literature search to support the variable selection argument and to set controls on raw data
quality. The details of variable selection are referenced in the earliest case study for a
scenario in Wisconsin (Chi, 2010).
Based on the Wisconsin dataset, our research team uses ordinary least squares (OLS)
regression, spatial lag regression, and spatial error regression to evaluate the relationship
between the index and natural amenities (Chi and Marcouiller, 2013). It is found that land
developability is positively associated with in-migration in Wisconsin, especially in remote
and rural areas, because of better natural amenities and controlling for other socioeconomic
and environmental factors.
With the use of county-level data from 2001 for the contiguous United States, this index
can be used to assess of urbanization, land use change, and deforestation (Clement et al.,
2015). Based on a two-way fixed-effects model, our research team finds that a county with
higher land developability in 2001 experiences a higher rate of severe deforestation between
2001 and 2006 (Clement et al., 2015).
We also compare the 2011 and 2011 county-level data with historical population
datasets (Chi and Ho, 2018) with the use of OLS regression, spatial lag regression, spatial
error regression, spatial error regression with lag dependence, and geographically weighed
regression. Our results show that decrease in land developability is associated with
population stress caused by population increases across the United States, and this
association with population stress can vary by location. Specifically, counties in the Midwest
and the traditional Deep South experience less population stress, while counties along the



Southeast Coast, Washington State, Northern Texas, and the Southwest are areas with
higher stress. This study also applies a differential Moran's *I* analysis that shows similar
findings as above.

In addition, recent study has also validated the use of the land developability index for

population projection (Chi and Wang, 2018). By using the 2011 land developability index, we
are also able to minimize percentage error for population projection from 2000 to 2010,
controlling for other factors such as socioeconomic statuses, crime rate, and transportation.

There is also a cross-validation from the public media. For example, a news reporter

compared the 2011 land developability index with the median home values in the 35 largest
cities in the United States. He found that a city with lower land developability has higher
housing prices than the others (Forbes, n.d.). Overall, the land developability index can be
practically used in demographic and policy-based assessments.

**4 Data availability**
The land developability index (Chi and Ho, 2020) generated by this work are publicly
available and can be downloaded at https://doi.org/10.7910/DVN/AMZMWH or
www.landdevelopability.org.

**5 Conclusions**
In this study, we presented an open-source dataset to measure land developability. This
dataset considered land vulnerability and development that can be restricted by both land
policy and geophysical limits. Particularly, we developed time-series data with a higher
flexibility considering the potential of land to be developed that can be used to identify areas
of spatiotemporal change. Our land developability directly addresses the issue that land
vulnerability and development cannot be simply quantified by land cover/use change caused
by population dynamics. Specifically, the land developability dataset has the ability to include



legal matters for a further identification of lands facing population stress, socioeconomic
burdens, and health risks. Based on the concept of "land developability", this spatial index is
aligned with policy-driven factors and land vulnerability to better reconcile developability
with socio-environmental justice. The first phrase of policy-driven land developability
mapping is implemented in estimating land use across the contiguous United States in 2001,
2006, and 2011. Multiscale data products for state-, county- and census-tract-levels are
provided from this estimation.

All the raw data for generating the land developability index come from remote sensing

images. Given the prevalence of remote sensing images across the world, the land
developability index could be produced for many regions. The remote sensing images do not
have to be in high resolution for most city or regional planning and policy purposes. Most
remote sensing images that are open to the public would be sufficient. The policy and
planning factors, though, need to be extracted from local context. The land developability
index could be modified for specific scenarios in other countries.

**Author contributions.**
GC initiated this investigation. GC designed the study. HH developed the model code and
performed the analysis. HH and GC prepared the paper.

**Competing interests.** The authors declare that they have no conflict of interest.

**Financial support.** This research was supported in part by the USDA National Institute of
Food and Agriculture and Multistate Research Project #PEN04623 (Accession #1013257), and
the Social Science Research Institute and the Institutes for Energy and the Environment of
the Pennsylvania State University.





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



**Figure Legends**
Figure 1. Web GIS interface for the 2001 land developability map at the county level.
Darker green indicates counties with higher land developability and lighter green indicates
counties with lower land developability.

Figure 2. Web GIS interface for the 2006 land developability map at the county level.
Darker green indicates counties with higher land developability and lighter green indicates
counties with lower land developability.


Figure 3. Web GIS interface for the 2011 land developability map at the county level.
Darker green indicates counties with higher land developability and lighter green indicates
counties with lower land developability.




Figure 1. Web GIS interface for the 2001 land developability map at the county level

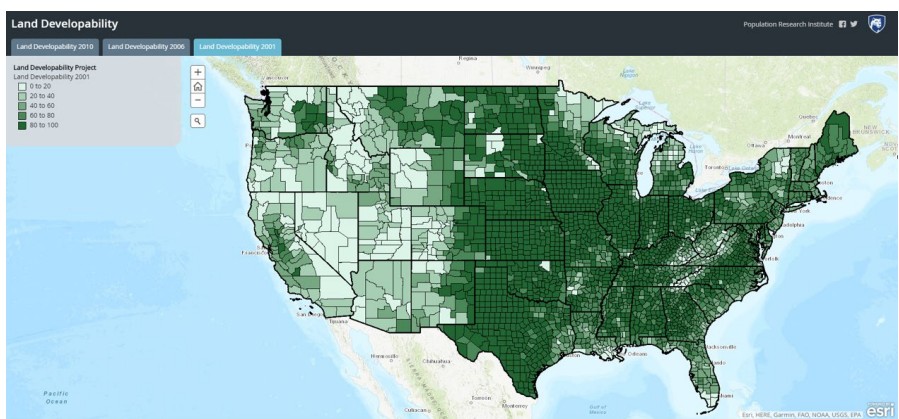







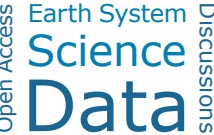

Figure 2. Web GIS interface for the 2006 land developability map at the county level

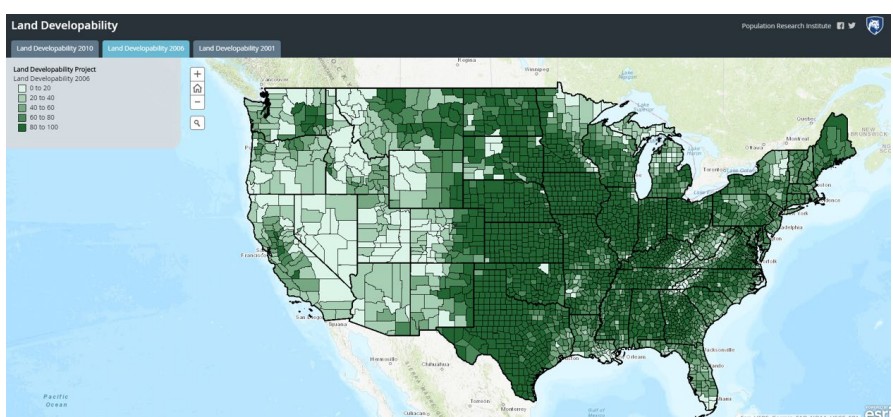








Figure 3. Web GIS interface for the 2011 land developability map at the county level

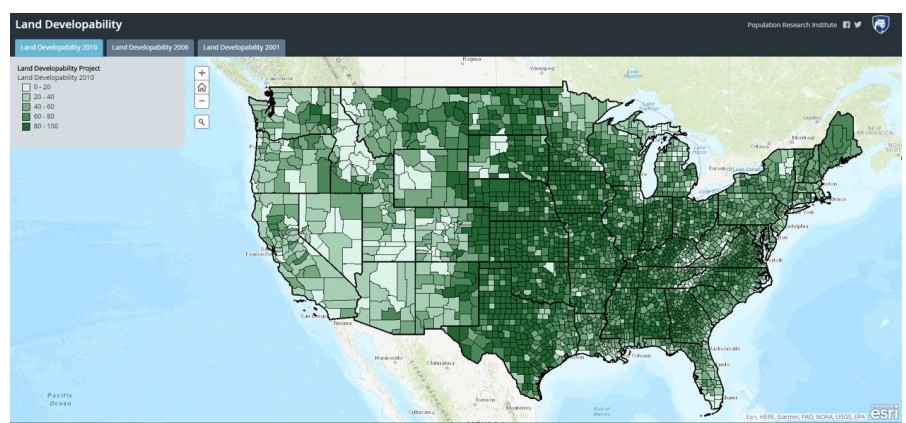
