# Peer review of "A Multiscale Spatial Dataset for Policy-Driven Land Developability across the"

_Earth System Science Data, 2020_

## Referee Comment (RC1) · Anonymous Referee #1 · 21 Apr 2020

This paper discusses the creation of a land availability index. For a paper that is meant to discuss the creation of a data set, it is poorly written and incomplete. The text is sloppy and has grammar problems. The introduction is overly generalized and makes references to things like "disasters predicted by Malthusian theory." This is just sloppy writing. I agree with the authors that's important to highlight the potential uses but their suggestions strike me as overly broad and not helpful.

On the index's usefulness: The data may be useful but it's not clear how the index was constructed. Since this is a paper about the construction of the data set much more care needs to be given how the exact details of the index and how, in theory, in could

be replicated.

On Uniqueness: Data sets like this have been created by economists over the last decade or so. So this data set does not strike me as particularly unique. See the following:

Saiz, A., 2010. The geographic determinants of housing supply. The Quarterly Journal of Economics, 125(3), pp.1253-1296.

Burchfield, M., Overman, H.G., Puga, D. and Turner, M.A., 2006. Causes of sprawl: A portrait from space. The Quarterly Journal of Economics, 121(2), pp.587-633.

Lutz and Sand 2017: paper and data: https://chandlerlutz.github.io/publication/land-unavailability/

On its quality: All I can say here is that I opened the shapefile for 2011 and it worked and seemed correct from what I can tell by simply visually inspecting the shapefile. But the authors are quite vague on how they created the index. So there's no way for me to know if what they did is accurate and correct, and how the index values may change with different assumptions or methods to create the index. All indexes should be validated by comparing different versions to see how well they correlate. Also there's nothing in the paper about how they validated their index or shown correlates with steepness or water area or anything to give the reader the confidence that the index is in fact usable data.

Personally I wouldn't use this data for county-level analysis. There's little in the paper that gives me confidence that it's a high-quality index. If the authors want it to be used in data work, they should go through all the elements in fine-grained detail and show the reader how the index was constructed and how its robust to small changes in the assumptions and then show some examples how the index correlates with the components and how the index correlates with other measures of land types and/or other variables that will help the reader have confidence in the index. They mention

they use the index is some regressions for Wisconsin but they ask the reader to take them on faith.

To be honest, the paper feels like it was rushed in order to get another publication.

---

## Referee Comment (RC2) · Anonymous Referee #2 · 13 May 2020

Review ESSD-2020-3, Land Development USA

This product elicits a very mixed response from this reviewer. Deficiencies seem to substantially outweigh positive benefits. Needs - at minimum - very substantial revision. Several typographic errors persist (the authors several times confuse 'phrase' with 'phase') but those can wait until authors fix larger problems.

Positive factors:
1.  I find (or know of) few or no similar compilation efforts, the authors having apparently produced a unique product;
2.  The authors help users understand some of their motivation;
3.  Data and tools seem accessible and useful (I can open shape files in QGIS, for example, avoiding license expenses associated with ESRI); and
4.  The authors have made a reasonable attempt at providing validation evidence.

Deficiencies:
1.  This reviewer questions whether the authors have identified and incorporated factors with the highest impact - at least in USA - on land developability;
2.  Agriculture - a politically, economically and geographically huge factor in USA land use, land availability and land developability where it severely restricts for example "residential, commercial, or other types of land development" (lines 108-109) -  seems largely ignored;
3.  One wonders whether the authors ignored agriculture or other developability factors due to lack of data (imagery), lack of open access to such data, or due to some impact weighting factor not clarified? Convenience from a remote sensing viewpoint - yes. Highest impact?;
4.  Related to prior comment, reader/user misses a data source table identifying each specific data source with exact source information, version, URL or DOI, etc. If authors expect readers to accept their claim of free and open remote sensing imagery as the basis of their index product, they need to provide exact explicit access information.
5.  Visually, largest changes in land use developability seem to occur 2006 to 2010, rather than 2001 to 2006. If quantitatively true, the authors provide neither mention, explanation, nor validation;
6.  Authors rely primarily on their own prior work (e.g. for Wisconsin) but seem to miss (or to neglect to explain the absence of) other prominent (in the land use community and in this journal) products such as night light data, census data, fertilizer use data, stream and ground water quality data, etc. Of products used here, water no water or slope > 20% seem historical and static (e.g. existing long before 2001), not changing much over their period of analysis, impervious surfaces (their surrogate for built-up area) seems weak and horizontal (did I see a different impermeable surface data set in ESSD, produced by and for hydrologists?) despite the fact that authors claim several times about the need to assess land use impacts vertically as well as horizontally, and tax exempt status (apparently binary in this analysis) misses a morass of benign to aggressive local-, county- or state-specific land tax policies. Which if any of their factors seem most likely to have changed over 2001 to 2010? Readers get no hint. Relates directly to uncertainty …;
7.  A user gains no sense of uncertainty, either of individual factors nor of the composite developability index / percentage. This reviewer suspects that authors can quantify developability in any USA county to no better than $\pm$ 20% in any time snapshot. Large uncertainties of specific snapshots then propagate to substantial uncertainties on any trend analysis. Authors owe users/readers an expert quantitative source level and compilation (index) level uncertainty analysis.

---

## Author Comment (AC1) · 1 Aug 2020

Response to reviewer 1

Comment 1: This paper discusses the creation of a land availability index. For a paper that is meant to discuss the creation of a data set, it is poorly written and incomplete. The text is sloppy and has grammar problems. The introduction is overly generalized and makes references to things like "disasters predicted by Malthusian theory." This is just sloppy writing. I agree with the authors that's important to highlight the potential uses but their suggestions strike me as overly broad and not helpful.

Response: Sorry for the inconvenience. We wrote it as a short communication, but we agree that more information should be added, as reviewer suggested. Therefore, we have extensively expanded the introduction section (Lines 35–118) to fulfill the requirements of a full-length article.

Comment 2: On the index's usefulness: The data may be useful but it's not clear how the index was constructed. Since this is a paper about the construction of the data set much more care needs to be given how the exact details of the index and how, in theory, in could be replicated.

Response: Thank you for the comments. We have now added more information regarding the theory and methods in the revised manuscript (Lines 91–118, 135–140, 142–150, 156–157, 173–175)

Comment 3: On Uniqueness: Data sets like this have been created by economists over the last decade or so. So this data set does not strike me as particularly unique. See the following: Saiz, A., 2010. The geographic determinants of housing supply. The Quarterly Journal of Economics, 125(3), pp.1253-1296. Burchfield, M., Overman, H.G., Puga, D. and Turner, M.A., 2006. Causes of sprawl: A portrait from space. The Quarterly Journal of Economics, 121(2), pp.587-633. Lutz and Sand 2017: paper and data: https://chandlerlutz.github.io/publication/landunavailability/

Response: Thank you for the suggestions. These publications are indeed good reads, and the dataset are very useful from an economic perspective. However, there is a major difference between our dataset and these data. While these data also considered natural and artificial environments as components, they did not consider the policy influences as well as elements of cultural heritage. This makes these datasets and our data differ regarding the intention of data development and potential applications. Specifically, our dataset is not an economic dataset that can be directly comparable with the data that the reviewer has suggested. More information has now been noted in the revised manuscript (Lines 91–118).
Comment 4: On its quality: All I can say here is that I opened the shapefile for 2011 and it worked and seemed correct from what I can tell by simply visually inspecting the shapefile. But the authors are quite vague on how they created the index. So there's no way forT me to know if what they did is accurate and correct, and how the index values may change with different assumptions or methods to create the index. All indexes should be validated by comparing different versions to see how well they correlate. Also there's nothing in the paper about how they validated their index or shown correlates with steepness or water area or anything to give the reader the confidence that the index is in fact usable data.

Response: Thank you for the suggestion. We have now added an additional analysis for spatial uncertainty, as suggested by both reviewers (Lines 235–240, 269–284, 350–355).

Comment 5: Personally I wouldn't use this data for county-level analysis. There's little in the paper that gives me confidence that it's a high-quality index. If the authors want it to be used in data work, they should go through all the elements in fine-grained detail and show the reader how the index was constructed and how its robust to small changes in the assumptions and then show some examples how the index correlates with the components and how the index correlates with other measures of land types and/or other variables that will help the reader have confidence in the index. They mention they use the index is some regressions for Wisconsin but they ask the reader to take them on faith. To be honest, the paper feels like it was rushed in order to get another publication.

Response: Thank you for the comments. We have now revised the manuscript with more details regarding theory.

Response to reviewer 2

Comment 1: This product elicits a very mixed response from this reviewer. Deficiencies seem to substantially outweigh positive benefits. Needs - at minimum - very substantial

revision.

Response: We have now made substantial revisions to incorporate the reviewers' comments and suggestions as much as possible. We believe the revised manuscript represents a much improved presentation of our work.

Comment 2: Several typographic errors persist (the authors several times confuse 'phrase' with 'phase') but those can wait until authors fix larger problems.

Response: Thank you for pointing out the typos. We have now proofread the manuscript more carefully.

Comment 3: Positive factors: 1. I find (or know of) few or no similar compilation efforts, the authors having apparently produced a unique product;

Response: Thank you for liking the contribution of our work.

Comment 4: 2. The authors help users understand some of their motivation;

Response: Thank you for the encouragement.

Comment 5: 3. Data and tools seem accessible and useful (I can open shape files in QGIS, for example, avoiding license expenses associated with ESRI);

Response: Thank you for pointing out that this data can be used as a quick tool on a GIS platform.

Comment 6: and 4. The authors have made a reasonable attempt at providing validation evidence.

Response: Thank you for the encouragement.

Comment 7: Deficiencies: 1. This reviewer questions whether the authors have identified and incorporated factors with the highest impact - at least in USA - on land developability;

Response: Thank you for the comments. We are sorry that we may have confused

readers. Our variables are picked based on land "not suitable" for land development. More information is provided in our comments below.

Comment 8: 2. Agriculture - a politically, economically and geographically huge factor in USA land use, land availability and land developability where it severely restricts for example "residential, commercial, or other types of land development" (lines 108-109) - seems largely ignored

Response: Thank you for the comments. We are sorry that we may have confused readers. Our variables are picked based on land "not suitable" for land development. As agricultural lands are suitable for development, we did not include them in the modeling. Consequently, our mapping shows that areas with more agricultural lands have higher developability. Our previous studies have also shown that our data can identify the relationship between agricultural activities and land development in the United States. More information has been added in Lines 322–327.

Comment 9: 3. One wonders whether the authors ignored agriculture or other developability factors due to lack of data (imagery), lack of open access to such data, or due to some impact weighting factor not clarified? Convenience from a remote sensing viewpoint - yes. Highest impact?;

Response: Thank you for the suggestion. Please refer to comment 8 regarding the process of data modeling.

Comment 10: 4. Related to prior comment, reader/user misses a data source table identifying each specific data source with exact source information, version, URL or DOI, etc. If authors expect readers to accept their claim of free and open remote sensing imagery as the basis of their index product, they need to provide exact explicit access information.

Response: Thanks for the suggestion. A new table (Table 1) has been added to the manuscript to provide this information.

[Figure]

Comment 11: 5. Visually, largest changes in land use developability seem to occur 2006 to 2010, rather than 2001 to 2006. If quantitatively true, the authors provide neither mention, explanation, nor validation;

Response: Thank you for the suggestion. The 2006 dataset is a test case for spatial uncertainty analysis. More information has been added to Lines 235–240; 269–284; 350–355.

Comment 12: 6. Authors rely primarily on their own prior work (e.g. for Wisconsin) but seem to miss (or to neglect to explain the absence of) other prominent (in the land use community and in this journal) products such as night light data, census data, fertilizer use data, stream and ground water quality data, etc. Of products used here, water no water or slope > 20% seem historical and static (e.g. existing long before 2001), not changing much over their period of analysis, impervious surfaces (their surrogate for built-up area) seems weak and horizontal (did I see a different impermeable surface data set in ESSD, produced by and for hydrologists?) despite the fact that authors claim several times about the need to assess land use impacts vertically as well as horizontally, and tax exempt status (apparently binary in this analysis) misses a morass of benign to aggressive local-, county- or state-specific land tax policies. Which if any of their factors seem most likely to have changed over 2001 to 2010? Readers get no hint. Relates directly to uncertainty . . .;

Response: Thank you for the suggestion. More quantitative information regarding the changes has been added to Lines 256–284. For the "validation" section, it has also been rewritten as a review of the application of the land developability index (Lines 286–319). As for the other limitations, they are addressed in Lines 328–355.

Comment 13: 7. A user gains no sense of uncertainty, either of individual factors nor of the composite developability index / percentage. This reviewer suspects that authors can quantify developability in any USA county to no better than + 20% in any time snapshot. Large uncertainties of specific snapshots then propagate to substantial

uncertainties on any trend analysis. Authors owe users/readers an expert quantitative source level and compilation (index) level uncertainty analysis.

Response: Thank you for the suggestion. We have now added an additional analysis for spatial uncertainty, as suggested by both reviewers (Lines 235–240; 269–284; 350–355).

——————————————————